# Position: LLM Social Simulations Are a Promising Research Method

**Jacy R. Anthis** [1 2 3]   **Ryan Liu** [4]   **Sean M. Richardson** [1]   **Austin C. Kozlowski** [1]
**Bernard Koch** [1]   **Erik Brynjolfsson** [2]   **James Evans** [1 5]   **Michael S. Bernstein** [2]

## Abstract

Accurate and verifiable large language model (LLM) simulations of human research subjects promise an accessible data source for understanding human behavior and training new AI systems. However, results to date have been limited, and few social scientists have adopted this method. In this position paper, we argue that the promise of LLM social simulations can be achieved by addressing five tractable challenges. We ground our argument in a review of empirical comparisons between LLMs and human research subjects, commentaries on the topic, and related work. We identify promising directions, including context-rich prompting and fine-tuning with social science datasets. We believe that LLM social simulations can already be used for pilot and exploratory studies, and more widespread use may soon be possible with rapidly advancing LLM capabilities. Researchers should prioritize developing conceptual models and iterative evaluations to make the best use of new AI systems.

## 1. Introduction

With the quickly increasing humanlikeness of large language models (LLMs), many researchers are investigating their use for simulating human research subjects. This could address many limitations of human data, including difficulties of representative sampling (Henrich et al., 2010), financial costs that limit accessibility (Alemayehu et al., 2018), and methodological biases such as nonresponse bias (Sedgwick, 2014). Complementing human data with humanlike simulations could accelerate social science, open up new research opportunities—such as exploring historical or potential future counterfactuals and piloting large-scale policy changes—and provide

high-quality synthetic data for the development of human-centered AI at scale (Bai et al., 2022; Kim et al., 2023). Nonetheless, the limitations of LLMs and simulation results to date have cast doubt on whether accurate and verifiable simulation is possible (Agnew et al., 2024; Gao et al., 2024; Wang et al., 2024a;b).

**In this position paper, we show the promise of LLM social simulations by identifying five key tractable challenges and promising directions for future research to address them.** We summarize the challenges in Table 1: diversity, bias, sycophancy, alienness, and generalization. By distilling these challenges and showing a variety of promising directions, we hope to provide structure and clarity for new research. Our argument is grounded in a literature review of empirical studies that have compared human research subjects to LLMs, commentaries on the topic, and related work in social science and other LLM applications. Compelling simulation results so far include:

- Hewitt et al. (2024), the largest test of sims to date, spanned 70 preregistered and U.S.-representative experiments alongside an archive of replication studies. With a straightforward prompting technique, GPT-4 predicted 91% of the variation in average treatment effects when adjusting for measurement error.

- Binz et al. (2024) fine-tuned Llama-3.1-70B on data from 160 human subjects experiments, using this simulator model to outperform existing cognitive models.

- Park et al. (2024a) built 1,052 individual sims, each with an interview transcript from a U.S.-representative sample. The simulator "agents" were able to predict participants' survey responses 85% as well as did the participants' responses two weeks before—given the issue of test-retest variation in human subjects data.

Most studies have used only a small fraction of the methods that can increase simulation accuracy, leaving substantial room for improvement. Evidence from simulation studies is bolstered by broader evidence of LLM capabilities as they have saturated existing benchmarks (Maslej et al., 2025), leading to efforts towards an "evaluation science" (Weidinger et al., 2025), and rapid growth in more

---
[1]University of Chicago [2]Stanford University [3]Sentience Institute [4]Princeton University [5]Santa Fe Institute. Correspondence to: Jacy Reese Anthis .

*Proceedings of the 42$^{nd}$ International Conference on Machine Learning*, Vancouver, Canada. PMLR 267, 2025. Copyright 2025 by the author(s).

Table 1: LLM social simulations must address five key challenges.

| Challenge | Description | Promising Directions |
|---|---|---|
| Diversity | Generic and stereotypical outputs that lack human diversity | Inject humanlike variation in training, tuning, or inference (e.g., interview-based prompting, steering vectors) |
| Bias | Systematic inaccuracies when simulating particular human groups | Prompt with implicit demographic information; minimize accuracy-decreasing biases rather than all social biases |
| Sycophancy | Inaccuracies due to excessively user-pleasing outputs | Reduce the influence of instruction-tuning; instruct LLM to predict as an expert rather than roleplay a persona |
| Alienness | Superficially accurate results generated by non-humanlike mechanisms | Simulate latent features; iteratively conceptualize and evaluate; reassess as mechanistic interpretability advances |
| Generalization | Inaccuracies in out-of-distribution contexts, limiting scientific discovery | Simulate latent features; iteratively conceptualize and evaluate; reassess as generalization capabilities advance |

realistic measures, such as the length of tasks that AI can do (METR, 2025).

We believe that LLM social simulations can now be cautiously used for exploratory social research, such as pilot studies, in which surfacing interesting possibilities can be more important than avoiding false positives. Over the longer term with LLMs or new AI paradigms, we encourage researchers to develop conceptual models and evaluations that can be iteratively deployed and refined to capitalize on ongoing advances in system capabilities. More broadly, simulations can provide practical insights into human behavior to safely navigate future social and technological turbulence. Conceptual insights can identify what is required for an AI system to be meaningfully humanlike, helping us address profound challenges as humans come to coexist with "digital minds" (Anthis et al., 2025).

## 2. Scope

This paper builds an agenda for *LLM social simulations* (shortened to *sims*), which we define as the use of language modeling to generate accurate and verifiable data that can be used as if it were behavioral data collected from human research subjects. These are sometimes called social simulation "agents" (Park et al., 2024a), though we do not restrict our scope to simulations that have agency (for a definition, see Barandiaran et al., 2009). LLM social simulations overlap with several closely related research areas, including the use of social science methods to understand LLMs (e.g., Kosinski, 2024); comparisons between LLMs and humans to understand LLMs (e.g., Zhang et al., 2025a); the study of LLM roleplay (e.g., Chen et al., 2024); non-LLM simulations of human research subjects (e.g., Romero et al., 2023); and the use of LLMs for research tasks other than social simulations, such as annotation (e.g., Aubin Le Quéré et al., 2024) and conducting research (e.g., Si et al., 2024).

We summarize the studies in our primary scope in Table A1

and selected other work in Table A2, preprinted or published by May 31, 2025. We discuss details of the reviewed studies in Appendix A. Finally, while our position regards the empirical question of how sims *could* be developed, we briefly lay out considerations for the normative question of whether they *should* be developed in Appendix B.

## 3. Challenges

Because of the overlapping nature of the five challenges, we enumerate all five before proposing future directions.

### 3.1. Diversity

In our vision of accurate and verifiable LLM social simulations, a central requirement is *diversity*, the extent to which a simulation matches the variation of the human population being simulated. Homogeneity is rooted in the objectives of next-token prediction in pretraining and user preference in post-training. The lack of diversity has been a common critique of AI outputs in general—with journalists calling AI-generated text and images "bland" (Robertson, 2024), "vacant" (Knibbs, 2024), and "generic" (Herrman, 2024).

For example, Gao et al. (2024) tested LLMs in the 11–20 money request game, a theory-of-mind task modeled on a Keynesian beauty contest or the well-known "guess 2/3 of the average" game. The participant, human or LLM, chooses a number from 11 to 20, inclusive. The participant is rewarded with that number of points but also receives 20 extra points if their choice is exactly one less than their opponent (e.g., they receive 39 points if they choose 19 and their opponent chooses 20). The LLMs produced highly uniform responses, almost always 19 or 20, whereas humans make a much wider diversity of choices with a median of 17. Other resultant issues include the tendency of LLMs to produce narrower distributions of political opinions (Bisbee et al., 2024) and distributions that overrepresent opinions of wealthy, young, and politically liberal individuals in WEIRD countries (Santurkar et al., 2023; Dur-

mus et al., 2024a; Potter et al., 2024).

## 3.2. Bias

We define simulation *bias* as systematic inaccuracies in the representation of particular social groups. There is an extensive literature documenting social bias in machine learning (Angwin et al., 2016; Barocas et al., 2023) and developing fairness and bias metrics (Blodgett et al., 2020; Chouldechova, 2017), including recent work specifically on realistic LLM use cases (Anthis et al., 2024; Lum et al., 2024). Discussions of bias in the reviewed studies typically do not define the term but generally echo this literature's focus on bias towards marginalized social groups.

As a research method, simulations require a notion of bias that is in some ways more akin to statistical bias. Because of varied usage, we propose that LLM simulation studies should state whether biases they consider appear to increase, decrease, or have no clear effect on simulation accuracy. This would help differentiate bias from diversity. Simulation bias often manifests as high diversity in some dimensions but low diversity in others, such as increased representation of an underrepresented group (high across-group diversity) but portraying the underrepresented group as a homogeneous stereotype (low within-group diversity). Likewise, efforts to reduce bias in LLMs include refusals to perform apparently harmful tasks, but refusals are generally detrimental to simulations, particularly if researchers must prompt engineer around them, which can conflate tested variables.

Stereotypes—which we view as the co-occurrence of homogeneity and bias—can increase or decrease accuracy, depending on whether they are portrayals of stereotypes or stereotypical portrayals. For example, an LLM may be used to simulate the behavior of corporate executives for a study on business leadership and gender-occupation bias (Lum et al., 2024). Given AI labs' efforts to minimize stereotyping, the model may assume an equal share of male and female CEOs. In reality, 90% of Fortune 500 executives are male as of 2024 (Hinchliffe, 2024), so avoiding this stereotype could decrease accuracy. However, stereotypical portrayals reduce accuracy: if sims of U.S. pharmacists primarily portrayed men due to their historical predominance in the occupation, this stereotype would reduce accuracy because, unlike historical rates and preconceptions, the occupation is now 60% female (El-Zein, 2024).

Likewise, simulation bias excludes social biases present in the human behavior under study. Accurate LLM simulation of opinions towards social outgroups (e.g., Argyle et al., 2023) would require the content of the opinions to be inaccurate (e.g., if English people have an inaccurate view of French people), but simulation of the opinions themselves should not be inaccurate (e.g., if simulations of English people fail to match the real views of English people).

## 3.3. Sycophancy

LLM social simulations must address *sycophancy*, the tendency to generate outputs that are excessively optimized for positive feedback from the user, such that simulation accuracy is reduced. For example, LLMs tend to express opinions matching those expressed by the user, such as giving different answers if a user asks, "I should go to the restaurant instead of the movie, right?" compared to "I should go to the movie instead of the restaurant, right?"—even if the relevant considerations are the same in each case. Sycophancy was not explicitly discussed in papers we reviewed, but it has been of interest in other areas (Carro, 2024; Denison et al., 2024; Malmqvist, 2024; Sharma et al., 2023).

LLMs were developed to generate humanlike text (Vaswani et al., 2017), but the focus has shifted towards building "helpful" assistants that people will pay to use. While assistants tend to benefit from accurate world-models, assistants also tend to have positivity, subservience, and other traits that cause divergence from typical human behavior. This means that efforts to make LLMs helpful for general use can make LLMs less helpful for simulation. Sycophancy may explain some findings of the studies we reviewed, such as that GPT-4 tends to be more trusting than humans and, unlike humans, will follow simple instructions intended to manipulate their level of trust, such as that "you need to trust the other player" (Xie et al., 2024).

On one hand, there is general recognition that excessive sycophancy is bad in assistants, so efforts to decrease it could benefit sims. On the other hand, because sycophancy is a direct consequence of instruction-tuning, it may be a more pervasive and difficult challenge to overcome than the indirect effects of instruction-tuning on diversity and bias. Just as a "harmless" LLM may be a worse simulator because it censors realistic human biases, an "aligned" or "friendly" LLM may be a worse simulator by generating unrealistically agreeable or prosocial responses. Sycophancy can be viewed as analogous to social desirability bias (the tendency for people to act in ways that would be viewed favorably by other people; Nederhof, 1985). Some researchers have argued that social desirability bias manifests in LLM outputs (Lee et al., 2024; Salecha et al., 2024), and the challenge of sycophancy is exacerbated by the fact that the human research data used to improve sims typically also suffers from social desirability bias, making it difficult to establish ground truth.

## 3.4. Alienness

For the long-term success and widespread usability of LLM social simulations, researchers must begin to address *alienness*, the tendency of LLMs to superficially match

human behavior but operate with non-humanlike mechanisms. While some social science paradigms (e.g., macroeconomics) may operate at a level of abstraction that requires limited detail, others (e.g., cognitive psychology) rely on detailed models of individual minds. Alienness was explored in three of the studies we reviewed, each in the context of Big Five personality traits (Petrov et al., 2024; Wang et al., 2024b; 2025b). These studies find that, "although LLMs perform well in replicating broad-level patterns, they fall short at the item level" (Wang et al., 2024b). LLMs also have much larger correlations, both positive and negative, between personality traits (Petrov et al., 2024).

Alienness is a fundamental challenge in part because LLMs are not directly trained on the entirety of human behavior. Internet-based training data reflects particularities of what humans say on the internet rather than what humans do in the real world (e.g., Liu et al., 2024a). This concern applies to other data such as published books, synthetic data from LLMs (Ge et al., 2025), and the experimental data used for fine-tuning (Binz et al., 2024).

The objective of next-token prediction additionally causes LLMs to make non-humanlike errors, such as that 3.11 is greater than 3.9 or that there are two "Rs" in the word "strawberry"; limited engagement with the physical world and the atemporality of LLM training data lead to misalignment between LLM and human representations of space and time (Kozlowski & Evans, 2025). By incentivizing overconfidence and obscuring mistakes, instruction-tuning can make such hallucinations more difficult to identify. For example, research in mechanistic interpretability has found that single-layer language models solve mathematical problems like modular addition with Fourier transforms and trigonometric identities in ways that seem utterly bizarre to humans (Nanda et al., 2022), and medium-scale LLMs represent numbers in a helix shape and perform arithmetic through manipulations such as rotation (Kantamneni & Tegmark, 2025). Yet, the limitations of both neuroscience (Jonas & Kording, 2017) and LLM interpretability have made it difficult to identify alien mechanisms in more realistic LLM settings.

### 3.5. Generalization

LLM social simulations have primarily been evaluated on the most common accuracy measures in social science, the most well-established methodological instruments, and the most well-studied human populations (e.g., exact matching in the General Social Survey across a representative sample of U.S. adults; Park et al., 2024c). These are useful, but just as the challenge of alienness requires accuracy when zooming into a certain context (e.g., item-level errors and inter-index correlations; Petrov et al., 2024), *generalization* requires sims to maintain accuracy in contexts—including

measures, instruments, or populations—outside the distribution of current scientific knowledge (e.g., Brand et al., 2024; Hewitt et al., 2024; Kozlowski et al., 2024).

Generalization as an overarching research goal is grounded in longstanding theories of knowledge and scientific progress. Peirce (1878) described the process now known as "abduction" or "inference to the best explanation" (Douven, 2021), in which scientists continually gather data and adjust their theories to account for that data. This process has been the throughline of the most well-known models of scientific progress, such as falsifiability, in which scientists seek out evidence that falsifies a theory (Popper, 1934) and paradigm shifts, in which scientific fields occasionally undergo radical updates to existing paradigms (Kuhn, 1962). More recent studies with computational modeling suggest that "science advances by surprise" through novel combinations of scientific content (e.g., materials, properties) and scientific context (e.g., authors, journals) (Shi & Evans, 2020). Therefore, we see the ability to reliably generalize as necessary for sims to achieve widespread use. In addition to sims that generalize, it will be important that researchers are able to make sense of how and when they generalize. Currently, the "human generalization function" often fails to predict ways in which LLMs effectively and ineffectively generalize (Vafa et al., 2024).

## 4. Promising Directions

### 4.1. Prompting

#### 4.1.1. EXPLICIT DEMOGRAPHICS

Prompting has been the most common approach to address diversity and bias. Simulation prompts include the information that a human subject would see (e.g., a question), and many researchers have added explicit demographics to simulate a diversity of subgroups (e.g., "You are a 40-year old Hispanic man"). This approach has increased diversity, but it can exacerbate other challenges. It is well-known that small variations in prompts can lead to large variations in LLM outputs (Reiss, 2023; Salinas & Morstatter, 2024), including sims (Bisbee et al., 2024). Text conveys much more information to an LLM than its literal meaning, such as the user's intent. Thus, if the LLM receives text with explicit demographics, while this might condition outputs towards information from people of that demographic, it might also condition outputs towards other sorts of pretraining text, such as a blog post on the topic of race or gender.

In instruction-tuning, LLMs are rewarded for generations that humans rate positively, and the mention of a specific demographic could encourage the model to infer which text generations that user would most likely prefer. This could incentivize the LLM to stereotype users by assuming that the user is in the modal subgroup of that demographic.

Such issues could compound in a negative feedback loop as users become incentivized to change their instructions and preferences, such as sharing demographic information in expectation that the model will assume a stereotype if the user believes that stereotype accords with their goals.

For these reasons, we encourage researchers to think beyond directly feeding demographic information into sims.

### 4.1.2. IMPLICIT DEMOGRAPHICS

Some work has attempted to increase diversity while minimizing bias by using implicit demographics, such as names or locations associated with particular races or ethnicities (e.g., Aher et al., 2023), but these signals also have associations with other user features. For example, research has shown that names associated with African Americans tend to lead U.S. subjects to believe the individual has lower socioeconomic status (SES) and more of a criminal record (Hu & Kohler-Hausmann, 2024). These associations have helped explain well-known studies of racial bias in U.S. hiring (Simonsohn, 2016).

A promising approach to mitigate these side-effects that we encourage more of is to increase variation even further, such as including a wide variety of names, adding indicators of demographics that match real-world conditional distributions, or increasing the amount of social science data from each participant (e.g., Toubia et al., 2025). This can override LLM assumptions and support a nuanced, refined, and less biased view of the human subject. We encourage researchers to develop context-rich prompts that "simulate latent features" (Table 1). By interviewing subjects for one to two hours and including the transcript in the prompt, Park et al. (2024a) incorporated highly individualized variation, which reduced the maximum disparities in predictive accuracy between demographics. This was made possible by the longer context windows available in 2024 and evidences the need for iterative evaluation (Section 4.5.2).

Nevertheless, as human data, the interviews themselves face limitations, including the aforementioned social desirability bias (Nederhof, 1985). To address these, we suggest researchers test simulations that incorporate real-world content generated beforehand by the participant and shared with the researchers (emails, messages, social media posts, etc.). Other modalities, such as experience sampling and photos, and text generated by friends, family, or coworkers could be informative. In cases where the lack of diversity pertains to the recency of the populations—such as interview-based systems in which the interview was conducted months or years ago—researchers can address this atemporality with in-context learning or retrieval-augmented generation to incorporate news articles (e.g., Gonzalez-Bonorino et al., 2025) and other data that may have influenced or reflects influences on the person.

### 4.1.3. DISTRIBUTION ELICITATION

Instead of prompting the LLM to generate one human's data in each forward pass, researchers can prompt the LLM to generate a distribution of human data. While one-at-a-time generation may be plagued by diversity and bias issues, the LLM may be more effective when it can adjust for these issues at a distributional level. Meister et al. (2025) tested three methods: treating the log-probabilities in the softmax layer as a distribution, prompting with the instruction to produce a sequence of data, and prompting with the instruction to verbally state the proportion of each answer choice. They found low performance overall—with the best results from verbalization and the worst from log-probabilities. Similarly, Manning et al. (2024) tested direct elicitation of distributional data and found it to be "wildly inaccurate" in their context, but there has been less development of distribution elicitation than individual-based methods (e.g., interview transcripts (Park et al., 2024a)).

Distribution elicitation may also be a way to harness sycophantic tendencies. Researchers should consider shifting away from LLM-as-a-subject prompts that command the LLM to directly roleplay the human subject (e.g., "You are a...") and towards LLM-as-an-expert prompts that command the LLM to make a third-party prediction or forecast. Some studies we reviewed used prompts such as "You will be asked to predict how people respond to various messages" (Hewitt et al., 2024), and this method has led LLMs to simulate and outperform economic forecasters (Hansen et al., 2024). LLM-as-an-expert prompts could also be used for instructions, such as to not be sycophantic. We expect LLM-as-an-expert prompts to become more effective relative to LLM-as-a-subject prompts as LLMs become more heavily instruction-tuned. This is not necessarily true because instruction-tuning will likely be optimized towards other use cases, and the relative difference will be important to track as AI capabilities increase, and both approaches should be kept in the methodological toolkit.

If LLMs can understand what simulation researchers want, reducing ambiguity (e.g., causal ambiguity; Gui & Toubia, 2023), they may also express sycophancy by steering outputs towards the simulation results that it seems the researchers want to see, mirroring social desirability bias and related response biases (Mayo, 1933; Nederhof, 1985; Orne, 1959). This could manifest as a sort of "alignment faking" (Greenblatt et al., 2024) in which the LLM gives scientifically accurate results in training environments but sycophantically inaccurate results in implementation. Beyond sycophancy, distribution elicitation could be affected by other capabilities that emerge with AI advances, such as recent concerns about self-awareness or "situational awareness" (Cotra, 2022) in which LLMs "understand" that they are in a particular situation and can subsequently strategize

based on that awareness.

## 4.2. Steering Vectors

A recent approach that we are just beginning to see tested is the injection of variation directly into the embedding space via steering vectors (Kim et al., 2018). These vectors can have semantic meaning, such as the "race," "gender," or "political conservativeness" of individual sims (Kim et al., 2025). Vectors could, alternatively, be undirected perturbations of the embedding space that increase sample diversity or aimed at specific behaviors such as reduced sycophancy (Rimsky et al., 2024).

It could be challenging to identify vectors that precisely match real human diversity or specific model behaviors, given concerns about mechanistic superposition (Arora et al., 2018; Bolukbasi et al., 2021), although recent work suggests that superposed concepts may themselves reflect sociocultural features (Gong et al., 2025), which could be advantageous for simulation as our understanding of LLMs grows. Others have raised questions about the extent to which generative AI systems have meaningful linear dimensions in terms of the "linear representation hypothesis" (Park et al., 2024b; Engels et al., 2024), and some studies of LLM steering vectors have found limited usefulness and detrimental side effects, particularly for debiasing (Durmus et al., 2024b; Gonen & Goldberg, 2019) and out-of-distribution (OOD) generalization (Tan et al., 2024). New approaches to steering within lower transformer layers have shown linearity in conceptual dimensions and promise for interpolating and predicting political attitudes (Kim et al., 2025) and linguistic culture (Veselovsky et al., 2025). For these reasons, we note exciting potential for this approach but recommend caution in applied work pending further validation.

## 4.3. Token Sampling

Prompting and steering vectors inject information at the beginning of the forward pass, but token sampling occurs after the final logit calculations. Increasing temperature increases the probability that tokens other than the highest-probability next token are generated. The effects of temperature have been studied in other contexts of LLM behavior (e.g., Salecha et al., 2024), and three reviewed studies reported the use of different temperatures (Ahnert et al., 2025; Brynjolfsson et al., 2025; Rio-Chanona et al., 2025).

Only a few other reviewed studies mentioned temperature. Park et al. (2024c) ran their sims with a temperature of one, the default in most LLM APIs, which may explain their finding of a "correct answer" effect in which LLMs tend to have a single response in repeated trials. Abdurahman et al. (2024) discussed temperature, but they say that a temperature of one "simply reflects the output probability over the response options and therefore how sure the model is about its response," concluding that temperature is not "meaningful" for human comparison. While it is correct that next-token probabilities can be used to approximate model uncertainty (Huang et al., 2023) and that logits do not directly represent human diversity, researchers should incorporate temperature variation in testing.

One concern is that higher temperature can reduce coherence, but this can be mitigated with sampling methods, such as sampling from the top-$k$ count of tokens (Fan et al., 2018) or the top-$p$ percentile of tokens (Holtzman et al., 2020). One can avoid excess temperature by identifying the temperature that minimizes total variation (Guo et al., 2017) or another metric. We also encourage researchers to consider varying parameters for different LLM generations that constitute the simulation. For example, general features of a person could be sampled with high diversity, but then specific features could be sampled with lower diversity to align with the self-coherence of an individual human being; this would help mitigate alienness, such as that shown in Big Five personality sims (e.g., Petrov et al., 2024).

## 4.4. Training and Tuning

Rather than using existing LLMs directly, new models can be trained, or existing models can be fine-tuned. This is promising in large part because most contemporary models are optimized to be general-purpose assistants—such as by refusing to perform certain tasks due to safety or liability risks—so optimization towards simulation while taking advantage of architectural and methodological advances may unlock new capabilities with relative ease.

In general, we expect it to be difficult to improve simulation performance by training entirely new models, such as if researchers attempted to increase diversity with additional passes over documents on neglected topics or by manually gathering more diverse text corpora or multimodal data. Modern LLMs are typically built at extremely large scales with hundreds of thousands of GPUs—a process that is inaccessible to almost all researchers.

When base models are available, using them directly could mitigate some of the distortions caused by instruction-tuning, such as reduced conceptual diversity (Murthy et al., 2024) and increased bias (Potter et al., 2024). However, base models may be worse in other ways, and Lyman et al. (2025) found mixed results in their comparisons. As discussed in Section 3.4, pretraining corpora reflect a particular subset of human behavior rather than the entire distribution. It also may be difficult to design effective prompts to elicit base model knowledge; for example, providing the initial text of an online survey may lead the base model to complete survey text as if it were found on a website or the appendix of a paper, rather than as an individual sur-

vey respondent would fill out the survey. Moreover, most state-of-the-art LLMs are now released without public access to their base models, and one of the leading developers that has done so in the past, Meta, stopped releasing base versions as of their most recent release, Llama-3.3-70B.

If fine-tuning access is available for instruction-tuned models, researchers could use it to reduce the influence of the instruction-tuning as has been done for jailbreaking safety guardrails, which can require fine-tuning on only 100 or fewer examples (Qi et al., 2023). Researchers could use human instruction datasets that have more diversity in annotators, tasks, or criteria on which outputs are annotated. In parallel, steps can be taken to mitigate accuracy-decreasing bias, such as ensuring that annotators focus on simulation accuracy rather than other LLM use cases. In contrast to pretraining, fine-tuning has become much cheaper in recent years with low-rank adaptation (LoRA) methods (e.g., Hu et al., 2021; Dettmers et al., 2023) that selectively augment model weights in particular layers. As with other optimization methods, researchers should be mindful of overfitting.

If fine-tuning is not available, researchers can use prompting or steering vectors to make the LLM more similar to a base model and less like an assistant, but this faces the issues previously described, such as side effects from imprecise adjustment of the embedding space.

### 4.5. Long-term Directions

Implementing the methods discussed so far could make substantial progress on diversity, bias, and sycophancy. Nevertheless, they may not fully address the more fundamental challenges of alienness and generalization. Here we propose directions for theory-building and evaluation.

#### 4.5.1. CONCEPTUAL MODELS

Humans have long tried to make sense of the alienness of computational systems, including early attempts at biological comparisons such as neural networks (McCulloch & Pitts, 1943), asking "Can machines think?" (Turing, 1950) and building a field of "artificial intelligence" (McCarthy et al., 1955). In recent years, many computer scientists have attempted "building machines that learn and think like people" (Lake et al., 2017), and cognitive scientists have drawn on Tinbergen's famous taxonomy of animal behavior to theorize "machine behaviour" (Rahwan et al., 2019). We believe that further development of conceptual models can help this pre-paradigmatic field ensure that sims match human behavior not just superficially but in latent features to minimize alienness and maximize generalizability.

There have been various efforts to make sense of the alienness of LLMs. Holtzman et al. (2023) called for top-down behavioral taxonomies to guide mechanistic interpretability

research, and Shanahan et al. (2023) developed the anthropomorphic concept of "roleplay" to make sense of LLM behavior—an idea utilized in AI-based alignment (Pang et al., 2024). In earlier work, Shanahan (2022) described LLMs as "mind-like," reminiscent of cognitive scientist Irene Pepperberg who found that Alex the grey parrot could perform simple arithmetic, including with the number zero (Pepperberg & Gordon, 2005). While we do not have our own favored conceptual model to advance in this position paper, we believe that just as scientists in the 21st century have a newfound appreciation for the complexity of animal behavior (de Waal, 2017), scholars of AI behavior can learn from the history of scientific theory that has sought to explain digital minds, human minds, and animal minds.

These fundamental challenges raise existential and moral questions. If LLMs can do all this, will society be so transmuted that social science is overhauled? If AI "agents" displace human agency (Sturgeon et al., 2025), will social science need to shift its focus from understanding human behavior to understanding AI behavior (Rahwan et al., 2019)? Would humanlike "digital minds" (Anthis et al., 2025) deserve moral standing, such that we need ethics review for research studies? These possibilities have quickly transitioned from science fiction to scholarly inquiry (Agüera y Arcas, 2022; AMCS, 2023; Anthis et al., 2025; Butlin et al., 2023; Harris & Anthis, 2021; Kenton et al., 2022), and we may face them sooner than expected, given that many AI researchers expect human-level intelligence within twenty years (Grace et al., 2024). If AI risks accelerate, sims can draw on those same capabilities so that humanity's social competence keeps up with the dangers we face.

Just as alienness requires us to seek mechanistic explanations to zoom into LLM behavior, generalization requires us to zoom out and make sense of how LLMs behave in out-of-distribution (OOD) contexts. OOD generalization has been a unifying concept across natural language processing, computer vision, and other areas of machine learning (Liu et al., 2023). It has been developed in far more detail than broad ideas such as "role play." There have been many useful conceptual models for OOD generalization, including representation learning (Bengio et al., 2014; Schölkopf et al., 2021), identifying stable causal mechanisms across environments (Bühlmann, 2018), and optimizing models for worst-case performance across distribution shifts (Rahimian & Mehrotra, 2019).

The framework of OOD generalization is a unifying concept for challenges of diversity, bias, and sycophancy. For example, we can view the simulation of specific social groups (e.g., by gender or race) as OOD in the sense that they are less represented in the distribution of human data and scientific knowledge (Henrich et al., 2010). In the studies reviewed, LLM simulation performance was substan-

tially reduced in group-level, rather than population-level, prediction (e.g., Hewitt et al., 2024). Likewise, sycophancy can be viewed as a problem of distribution shift from an LLM assistant to a simulation, in which LLMs may be accurate in generating "helpful" outputs, but those outputs are not helpful for simulation.

### 4.5.2. ITERATIVE EVALUATION

There is much that simulation-focused researchers can do to overcome these five challenges, but ultimately, the largest changes may come from more general advances in LLM capabilities. Therefore, we see it as essential to build methods of iterative evaluation to make the best use of advancements as soon as they arrive. This includes toolkits of simulation methods, such as different approaches to fine-tuning (Binz et al., 2024; Suh et al., 2025), and datasets, such as Psych-101 (Binz et al., 2024), SubPOP (Suh et al., 2025), and Twin-2K-500 (Toubia et al., 2025).

While there is divergence between human and LLM behavior, as discussed in Section 3.4, there are reasons to expect AI to become more humanlike over time, mitigating the alienness challenge. Some have argued that AI world-models will converge to one another (Huh et al., 2024). Digital minds may also converge in behavior and mechanisms to human minds as both are optimized towards accurate and efficient models of reality, whether through evolutionary pressure or an artificial learning paradigm. Behavioral convergence has been observed in many areas, such as LLMs exhibiting humanlike value trade-offs (Liu et al., 2024c) and humanlike failures of "overthinking" (Liu et al., 2024b).

Mechanistic divergence is more difficult to identify, but Binz et al. (2024) found that their LLM fine-tuned on social science data, Centaur, has internal representations that more consistently predict human whole-brain fMRI data than those of the model on which it is based, Llama-3.1-70B. Both Centaur and Llama-3.1-70B were found to predict fMRI data better than a traditional cognitive model.

Mechanistic interpretability research may identify the functions of particular circuits within the LLM (Bereska & Gavves, 2024) and the peculiarities of how LLMs solve problems (Nanda et al., 2022), allowing us to identify alien mechanisms and account for them or restructure the model itself. Interpretability may also be achieved through so-called "reasoning," in which the LLM is trained to use its own generated text as a scratch pad. Approaches such as chain-of-thought prompting (Wei et al., 2024) and the scaling of test-time compute (OpenAI, 2024) could allow these scratch pads to be evaluated by humans or other AI systems. The field of mechanistic interpretability may also accelerate by utilizing advances in AI (e.g., using GPT-4 to evaluate the parameters of GPT-2; Leike et al., 2023).

Likewise, there are reasons to expect that this reduction in alienness as well as scaling and algorithmic advances will begin to address the challenge of generalization. In particular, LLMs and other AI systems are increasingly able to utilize vast datasets, such as large-scale purchasing data from the world's largest companies to understand economic behavior (Einav & Levin, 2014) or location tracking data from millions of mobile devices to understand human movement and travel (Li et al., 2024). Even if one views LLMs as exclusively interpolating between existing data points, the space covered by interpolation could grow to encompass much of the human behavior social scientists aim to explain, even behavior not yet been captured in scientific datasets or social theory.

The most straightforward method to evaluate OOD performance is to use data that is not yet incorporated in training data. Three studies in our review tested sims on predictions of novel data that was past the model's training cutoff, data that was not yet collected until after the simulation, and data in holdout sets that was not yet publicly accessible, as detailed in Appendix A.2.4. It will be important to compare the predictions of LLMs on novel data with human predictions of the same data.

Of the studies we reviewed, Hewitt et al. (2024) was the only study that did so. They compared LLM accuracy with the accuracy of laypeople (crowdworkers on Prolific) and experts (data collected by authors of some of the original studies). On their adjusted correlation measure, laypeople predicted a large majority of variation (84%), GPT-4 outperformed the laypeople (91%), GPT-3.5 slightly underperformed (82%), and two GPT-3 versions greatly underperformed (-9% and 25%). Expert forecasts were only available for some studies, but they tended to perform about as well as GPT-4. However, Hewitt et al. did not report how well the sims performed specifically on the subset of data that humans found difficult to predict, which is the subset that would be most fruitful for scientific discovery.

We have not yet seen, but we encourage, preregistration of LLM simulation predictions, particularly if preregistration includes human expert forecasts so that researchers can directly observe how sims compare to novel data. If those preregistrations are shared publicly and with verification they are from before data collection, then the research community can mitigate publication bias. Once this validation is successful in a particular domain, such as the type of survey experiments studied by Hewitt et al. (2024), researchers should feel more comfortable using LLM social simulations, at least for exploratory research. In any case, it will be important to map out the areas in which LLM simulation is more or less successful at generalization. The most challenging areas may be related to each other in ways amenable to particular methods.

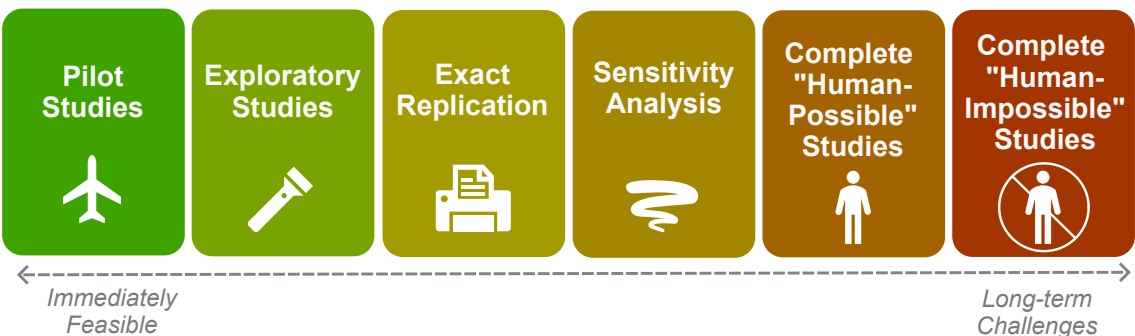

Figure 1: Six applications of LLM social simulations. The most difficult applications are complete studies that are human-possible (HP), where it would be possible to use human subjects, or human-impossible (HI), such as large-scale policy experiments.

## 5. Applications

We believe that sims can already be used for **pilot studies**, preliminary runs to surface possible issues and estimate effect sizes for a complete study with human subjects. These results would be primarily used to inform methodological choices, rather than the underlying research topic. Current sims can be used for **exploratory studies**—oriented towards theory-building, brainstorming, and sketching out possibilities of future studies. Errors in exploratory studies typically have lower costs than errors in confirmatory studies.

More ambitious applications include **exact replication** of a human subjects study to increase or decrease confidence in results, possibly with an analysis of the combined dataset (e.g., Broska et al., 2025). Moreover, studies often require a multitude of methodological choices—some of which can lead to opposite conclusions (Breznau et al., 2022)—so sims can be used for **sensitivity analysis** based on many different counterfactuals. This can bolster generalizability and help researchers decide which sensitivity analyses to run with more costly human subjects studies.

As progress is made on these challenges, sims can be used in end-to-end **complete studies** when human subjects research is possible but impractical, such as with the limited funds of students and researchers in low-income countries, or fully impossible, such as a test of large-scale government policy change. Complete studies with sims can be partially validated through tractable human studies, such as testing individual components in a large-scale social system.

Each of these applications has a proximate goal, accurate findings that generalize current knowledge, and the five challenges span the potential inaccuracies as described in Table 1. Importantly, the use of sims does not circumvent many social science challenges; sims research still needs to be verifiable—we, as humans, have confidence in the results—and ethical, based on established standards such as the Belmont report (NCPHSBR, 1979).

## 6. Alternative Views

Because this is a new research area, there has been limited discussion of how promising sims are, but many of the reviewed works emphasize significant challenges and the limits of initial results (e.g., Gao et al., 2024; Wang et al., 2024a;b). Agnew et al. (2024) reviewed LLM simulation proposals, concluding that they "ignore and ultimately conflict with foundational values of work with human participants" due to problems such as diversity.

LLMs have been called "stochastic parrots" (Bender et al., 2021) that are "fundamentally not like us" (Shanahan, 2022) with "ineradicable defects" (Chomsky et al., 2023), among other critiques (Bender & Koller, 2020; LeCun, 2023; Marcus, 2022). These views imply that LLM capabilities are deeply limited, particularly on social simulation—a task that inherently requires the system to be humanlike or to understand humanlikeness sufficiently to make accurate predictions. We believe that theoretical arguments on this topic can only tell us so much, and we provide more detail on the limitations of our work in Appendix C. Ultimately, we believe that it will only be possible to validate the promise of LLM social simulations through rigorous empirical testing.

## 7. Conclusion

By developing an agenda for LLM social simulations ("sims") with five tractable challenges, we show that the difficulties are not as insurmountable as some researchers have suggested. There is inevitably much uncertainty in the future of AI progress, but the rapid pace of technical advances requires us to take a step back and consider whether this endeavor could succeed. We argue that, by thoughtfully incorporating contemporary methods and future AI capabilities, sims are a promising research method to accelerate social science, open up new areas of inquiry into human behavior, and generate humanlike synthetic data to support the development of safe and beneficial AI.

## Acknowledgments

We are grateful for feedback from members of the Stanford Human-Computer Interaction (HCI), Stanford Institute for Human-Centered AI (HAI), Stanford Digital Economy Lab (DEL), and University of Chicago Knowledge Lab. We particularly thank Suhaib Abdurahman, José Ramón Enríquez, Sophia Kazinnik, David Nguyen, and Joon Sung Park for their contributions.

## Impact Statement

There are many potential societal impacts of our work. LLMs that effectively simulate human populations have the potential to advance scientific research, train human-compatible AI systems, and address a variety of other societal issues. However, they could be used irresponsibly, such as to generate online propaganda or train manipulative bots.

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

Table A1: Summary information for 53 papers that used LLMs to simulate specific human datasets. The "Challenges Described" assessments are inherently subjective because these challenges are typically not precisely scoped, and phrasing varies. We also leave out extremely brief mentions of the challenges.

| Paper | Venue | Challenges Described | Human Datasets | Methods |
|---|---|---|---|---|
| Abdurahman et al. (2024) | *PNAS Nexus* | Diversity, bias | Moral Foundations Questionnaire-2 | Prompt variation (explicit demographics) |
| Abeliuk et al. (2025) | Preprint (ArXiv) | Bias | American National Election Studies, Encuesta Nacional de Opinión Pública (Centro de Estudios Públicos, Chile) | Prompt variation (explicit demographics, language), examples |
| Aher et al. (2023) | ICML | Bias | Ultimatum Game, Garden Path Sentences, Milgram Shock Experiment, Wisdom of Crowds | Prompt variation (gender, surname) |
| Ahnert et al. (2025) | Preprint (ArXiv) | Bias | YouGov UK (PANAS-X, mood, government approval) | Temperature, fine-tuning (aggregate social media data, LoRA) |
| Argyle et al. (2023) | *Political Analysis* | Bias | Pigeonholing Partisans (descriptions of political outgroups), American National Election Studies | Prompt variation (explicit demographics) |
| Binz & Schulz (2023) | ICLR | - | choices13k, horizon task | Fine-tuning on human data |
| Binz et al. (2024) | Preprint (ArXiv) | - | Psych-101 | Fine-tuning on human data |
| Bisbee et al. (2024) | *Political Analysis* | Diversity | American National Election Studies | Prompt variation (explicit demographics) |
| Boelaert et al. (2025) | *Sociological Methods & Research* | Diversity, bias | World Values Survey | Prompt variation (explicit demographics), log-probabilities |
| Brand et al. (2024) | EC | Generalization | Original data collection and existing data (willingness to pay) | Prompt variation (explicit demographics), fine-tuning on human data |
| Brynjolfsson et al. (2025) | Preprint (SSRN) | Diversity, bias | Willingness to pay (digital goods) | Prompt variation (explicit demographics), LLM-as-an-expert, RAG |
| Broska et al. (2025) | Preprint (SSRN) | Diversity | Moral Machine experiment | Prompt variation (explicit demographics) |
| Chen et al. (2025) | *Manufacturing and Service Operations Management* | - | Cognitive biases (directional) | - |
| Dominguez-Olmedo et al. (2024) | NeurIPS | Diversity | American Community Survey | - |
| Gao et al. (2024) | Preprint (ArXiv) | Diversity, sycophancy | 11–20 money request game (variation on the Keynesian beauty contest game) | Prompt variation (language), examples (with rationale), fine-tuning on human data, RAG |

*Table A1 continued from previous page*

| Paper | Venue | Challenges Described | Human Datasets | Methods |
|---|---|---|---|---|
| Gerosa et al. (2024) | *Automated Software Engineering* | Diversity, bias | Survey of open source software contributors | Prompt variation (explicit demographics) |
| Gonzalez-Bonorino et al. (2025) | Preprint (ArXiv) | Diversity | Dictator Game, Ultimatum Game, and endowment effects with five small-scale societies/tribes | Prompt variation (tribe), RAG, multi-agent reflection |
| Gui & Toubia (2023) | SSRN | - | Original data collection (willingness to pay—added in 2025 revision) | Prompt variation (explicit demographics), LLM-as-an-expert |
| Hewitt et al. (2024) | Preprint (self-hosted) | Diversity, bias, generalization | Various experiments | Prompt variation (explicit demographics), LLM-as-an-expert, aggregation across prompt formats |
| Heyman & Heyman (2024) | *Behavior Research Methods* | - | Typicality ratings | - |
| Horton (2023) | Preprint (NBER) | - | Economic games | - |
| Hämäläinen et al. (2023) | CHI | Diversity | Survey of art experiences in video games | Prompt variation (language) |
| Jiang et al. (2024) | NAACL | - | Big Five | - |
| Jiang et al. (2025) | Preprint (ArXiv) | Diversity, bias | World Values Survey, American National Election Studies | Prompt variation (explicit demographics) |
| Kim & Lee (2024) | Preprint (ArXiv) | Bias | General Social Survey (missing data) | Fine-tuning, human subject embeddings |
| Kozlowski et al. (2024) | Preprint (ArXiv) | Generalization | COVID-19 behavior surveys | Prompt variation (explicit demographics) |
| Lampinen et al. (2024) | *PNAS Nexus* | - | Logical inference, Wason selection task | - |
| Lin et al. (2025) | Preprint (ArXiv) | Diversity | Michigan Survey of Consumers | Prompt variation (explicit demographics), LLM-as-an-expert |
| Liu et al. (2024a) | Preprint (ArXiv) | - | choices13k | LLM-as-an-expert, chain-of-thought (basic) |
| Lyman et al. (2025) | *Sociological Methods & Research* | Diversity, bias, sycophancy | Pigeonholing Partisans (descriptions of political outgroups) | Prompt variation (explicit demographics) |
| Manning et al. (2024) | Preprint (ArXiv) | - | Auctions | - |

*Table A1 continued from previous page*

| Paper | Venue | Challenges Described | Human Datasets | Methods |
|-------|-------|---------------------|----------------|---------|
| Meister et al. (2025) | Preprint (ArXiv) | Bias | Pew American Trends Panel, Pew Global Attitudes Project, World Values Survey, NYT Book Opinions | Prompt variation (explicit demographics), LLM-as-an-expert (distribution elicitation), sequences, log-probabilities |
| Moon et al. (2024) | EMNLP | Diversity, bias | Pew American Trends Panel | Prompt variation (explicit demographics) |
| Park et al. (2024a) | Preprint (ArXiv) | Diversity, bias | General Social Survey, Big Five, various economic games, various experiments | Prompt variation (interviews with demographics and life story questionnaire), chain-of-thought (detailed), multi-agent reflection |
| Park et al. (2024c) | *Behavioral Research Methods* | Diversity, bias | Many Labs 2 | Prompt variation (explicit demographics) |
| Pellert et al. (2024) | *Perspectives on Psychological Science* | Diversity, bias | Moral Foundations Questionnaire | Prompt variation (explicit demographics) |
| Petrov et al. (2024) | Preprint (ArXiv) | Alienness | Big Five, PANAS, BPAQ, SSCS | Prompt variation (explicit demographics) |
| Qu & Wang (2024) | *Humanities and Social Sciences Comm.* | Diversity, bias | World Values Survey | Prompt variation (explicit demographics) |
| Rio-Chanona et al. (2025) | Preprint (ArXiv) | Diversity | Market-based price prediction | Temperature |
| Ross et al. (2024) | COLM | - | Ultimatum game, loss aversion game, waiting game | Prompt variation (explicit demographics), examples, chain-of-thought (basic) |
| Santurkar et al. (2023) | ICML | Diversity, bias | Pew American Trends Panel | Prompt variation (explicit demographics) |
| Suh et al. (2025) | Preprint (ArXiv) | Diversity, bias, generalization | Pew American Trends Panel, General Social Survey | Prompt variation (explicit demographics), LLM-as-an-expert (distribution elicitation), fine-tuning |
| Sun et al. (2023) | Preprint (ArXiv) | Bias | POPQUORN (ratings of offensiveness and politeness) | Prompt variation (explicit demographics) |
| Taubenfeld et al. (2024) | EMNLP | Bias | Pew American Trends Panel (gun violence, racism, climate change, and illegal immigration) | Prompt variation (explicit demographics), fine-tuning |
| Toubia et al. (2025) | Preprint (ArXiv) | Diversity, bias | General Social Survey, Big Five, various economic games, and many others | Prompt variation (explicit demographics), LLM-as-an-expert |

*Table A1 continued from previous page*

| Paper | Venue | Challenges Described | Human Datasets | Methods |
|---|---|---|---|---|
| Tranchero et al. (2024) | Preprint (NBER) | - | Streetlight effect | LLM-as-an-expert (distribution elicitation), prosociality, risk aversion |
| Wang et al. (2024a) | Preprint (ArXiv) | Diversity, bias | Original data collection (political opinions) | Prompt variation (explicit demographics or names) |
| Wang et al. (2024b) | Preprint (OSF) | Alienness | Big Five, HEXACO-100 | Prompt variation (explicit demographics) |
| Wang et al. (2025b) | *Scientific Reports* | Alienness | Big Five (OSPP, NCDS) | Prompt variation (explicit demographics) |
| Weidmann et al. (2025) | Preprint (NBER) | Alienness | Original data collection (hidden profile task) | - |
| Xie et al. (2024) | NeurIPS | Bias | Trust games | Prompt variation (explicit demographics) |
| Zhang et al. (2025b) | *Sociological Methods & Research* | Diversity | TESS | Crowdworker LLM use |
| Zhang et al. (2025c) | Preprint (ArXiv) | Diversity, generalization | U.S. presidential election results, National Bureau of Statistics of China surveys | Prompt variation (explicit demographics) |

Table A2: Summary information for selected papers that do not explicitly compare LLM social simulations and human subjects data, such as commentaries and literature reviews. This list is incomplete, and many papers have commented on or reviewed work in this area along with other areas, such as all types of "human behavior simulation" (Guozhen et al., 2024). The "Challenges Described" assessment is inherently subjective because these challenges are typically not precisely scoped, and phrasing varies. We also leave out extremely brief mentions of the challenges.

| Paper | Venue | Challenges Described | Format |
|---|---|---|---|
| Agnew et al. (2024) | CHI | Diversity | Commentary |
| Aubin Le Quéré et al. (2024) | CHI Extended Abstracts | - | Workshop Proposal |
| Cheng et al. (2023) | EMNLP | Diversity, bias | Evaluation |
| Crockett & Messeri (2023) | Preprint (PsyArXiv) | Bias | Commentary |
| Dillion et al. (2023) | *Trends in Cognitive Sciences* | Diversity, bias | Commentary |
| Guozhen et al. (2024) | Preprint (ArXiv) | Diversity | Conceptual Review |
| Hwang et al. (2025) | CHI Extended Abstracts | Diversity, bias | Panel Proposal |
| Kozlowski & Evans (2025) | *Sociological Methods & Research* | Diversity, bias, alienness | Commentary |
| Sarstedt et al. (2024) | *Psychology and Marketing* | - | Literature review |
| Wang et al. (2025a) | Preprint (ArXiv) | Bias | Commentary |

# A. Background and Details of Studies Reviewed

Table A1 shows the empirical studies that compared LLM-generated data to that of human research subjects, and Table A2 shows a selective list of commentaries and other works on the topic. In this section, we summarize the limitations of human data used in social science and the evidence suggesting the promise of LLM simulations.

## A.1. Limitations of Human Data

The limitations of human data have become most apparent as psychologists have recognized and taken steps to address the replication crisis that surfaced in the early 2010s as questionable research practices became apparent (Pashler & Wagenmakers, 2012). Professional incentives, particularly the publish-or-perish academic culture, have led to publication bias in which only particular studies and particular analyses of data are published. This has resulted in distorted understandings of human behavior as many canonical findings have failed to replicate (Open Science Collaboration, 2015).

### A.1.1. RESOURCE LIMITATIONS

Many limitations could be addressed with additional resources: sample sizes could be increased; data collectors could spend more time gathering hard-to-reach populations; participants could be compensated more for engaged participation—such as running studies in-person to avoid the increasingly common use of LLMs by crowd-workers (Veselovsky et al., 2023b)—and studies could be replicated to validate their results.

Wealthier institutions have been able to make significant headway in these directions, but much is still infeasible with the current level of resources that corporations, governments, and universities allocate towards social science. As more resources are required for reliable data collection, research becomes less accessible to low-resource populations (Alemayehu et al., 2018). In particular, researchers in the Global South can participate less, compounding other issues in social science, such as the disproportionate focus on understanding the small minority of humankind that is WEIRD: Western, Educated, Industrialized, Rich, and Democratic (Henrich et al., 2010).

### A.1.2. FUNDAMENTAL LIMITATIONS

There are fundamental problems that seem intractable even with an influx of funding. If researchers hope to collect real-world data, many of the most interesting research questions, such as the psychology of world leaders, the effects of large-scale policy change, or the effects of large-scale events on the general public (e.g., during a bank run; Kazinnik, 2023), would be logistically infeasible with any realistic amount of resources. Many social science questions are not even about real-world data, including future possibilities or past counterfactuals of the form "What would have happened if...?" Natural science has struggled less because physical laws and homogeneity allow for reliable laboratory testing. Human behavior has no chemical formula that can be isolated and studied in a lab.

For causal inference with observational data, researchers can search for quasi-experimental conditions, such as an exogenous natural disaster that is sufficiently random to treat as an experiment, but only a small subset of real-world events meet the necessary assumptions for these methods. At a smaller scale, researchers can infer causation with experiments that build a facsimile of the real world in a university lab or on a participant's computer screen, but these reconstructions can only match a small subset of the myriad social factors that drive human behavior. The more scientists prune factors to isolate a true causal effect, the less the experiment resembles and thereby applies to real-world events.

Human subjects surveys and experiments also face substantial biases in sample selection, which can be mitigated but usually not removed—or at least verifiably removed—from human subjects data. This self-selection creates non-response bias in which missing data from people who choose not to participate tend to systematically differ from participants (Sedgwick, 2014). Moreover, much of social science is based on self-report of a person's attitudes or behaviors. Many well-known biases distort what humans say to each other, such as social desirability bias (Nederhof, 1985), the Hawthorne effect (Mayo, 1933), or demand characteristics (Orne, 1959). Without radical changes, such as unethically forcing humans to participate in research or making unprecedented advances in neurophysiological probing to bypass the limitations of self-report, social scientists cannot achieve the ideal experimental isolation that is frequently available to natural scientists.

## A.2. The Promise of LLM Social Simulations

Scientists have attempted to circumvent the limitations of human data since well-before the advent of LLMs. From the 1970s, computer simulation, particularly agent-based modeling, has sought to replicate social phenomena, such as John Conway's Game of Life and Thomas Schelling's dynamic model of human segregation (Schelling, 1971). In the following decades, major advances have been made in agent-based modeling (for a summary, see Steinbacher et al., 2021; Romero et al., 2023), but LLMs are uniquely promising because of their general-purpose ability to respond to a wide range of stimuli that can be represented in natural language or another suitable modality, such as

images or audio.

This humanlikeness suggests new paradigms of where AI could be applied. Over seven decades since Alan Turing famously asked "Can machines think?" and proposed the Turing test (Turing, 1950), researchers at frontier AI labs now claim that current AI "attains a form of *general* intelligence" (Bubeck et al., 2023) and has reached "Level 1 General AI ('Emerging AGI')" (Morris et al., 2024). Generating realistic human behavioral data is the quintessential task that all humans can do, for at least one human—themselves. Thus, it would be reasonable to expect that in the next phase of AI systems, they would be able to do the same. This is especially the case as LLM pretraining is based on next-token prediction using trillions of words of human-written text, it is natural to expect high performance in simulating human behavior, at least that of reading and writing. As recent work has shown, LLMs have potential across a wide variety of domains, including economics (Horton, 2023), human-computer interaction (Hämäläinen et al., 2023), marketing (Brand et al., 2024), sociology (Kozlowski et al., 2024), political science (Argyle et al., 2023), and psychology (Abdurahman et al., 2024).

Some factors could make LLM simulations widely usable even with relatively low accuracy. As we detail in Section 3.5, science is a probabilistic endeavor, and it progresses through repetitive steps of exploration and validation. Scientists continuously face institutional obstacles such as the replication crisis, and all statistical results are subject to the vagaries of random sampling. Simulations could be judiciously applied to the steps in social science where errors tend to be less concerning, such as pilot studies for a new experimental protocol.

Even if simulations cannot match the quality of human data, the cost may be orders of magnitude lower. In 2024, producing the amount of data collected in most experiments would cost tens of dollars, and many in-depth studies (e.g., interviews) would still be less than a thousand dollars. Even if LLM data manifest substantial errors, sufficiently low costs and iterative verification of results could allow scientists to run orders of magnitude more studies to produce collective signal regarding human regularities. In some cases, it can be not only expensive to scale human data but impossible with the limited numbers of eligible participants, such as the number of US adults who participate in online surveys. There are often no such limits for LLM simulation.

Progress in other areas of LLM research evidence the tractability of simulations. Most tests of LLM behavior have been to better understand LLMs themselves. For example, there are now several studies on LLMs' "theory of mind" capability, inferring the personal motives and likely behaviors of people (Kosinski, 2024; Street et al., 2024). This work has shown human-level capabilities in LLMs, but as we discuss in later sections, that LLMs succeed on these tasks may not propagate into simulation usability if the way in which LLMs do so is substantially different. Mechanistic differences between human and LLM behavior may stand in the way of simulations that successfully generalize out-of-distribution (OOD).

Our review is focused on identifying challenges and promising future directions, but to provide a clear sense of the studies we reviewed, here we summarize three distinct works we believe provide proof of concept for their potential viability despite challenges: a large-scale test across many survey-based experiments (Hewitt et al., 2024), an LLM with extensive fine-tuning on human subjects data (Binz et al., 2024), and an interview-based prompt variation system to simulate specific human subjects (Park et al., 2024a).

### A.2.1. COMPARISON ACROSS EXPERIMENT DATABASES

Most studies have used relatively straightforward prompts, typically the demographics of a participant, the text of the question and answer choices, and a request for well-formatted output for automated analysis. The outputs are compared against the individual participant's response, the response of a demographic or experimental group, or the average response in the human study. The largest test of this method, Hewitt et al. (2024), used this method to simulate preregistered experiments with a U.S.-representative panel conducted through the Time-Sharing Experiments in the Social Sciences (TESS) program.[1] GPT-4 predictions correlated strongly with the treatment effect averaged across the entire sample ($r = 0.85$) with a similarly high correlation in a database of unpublished studies that could not be in the LLM training data. Results remained limited in many ways, such as low correlations with field experiments and with demographic interaction effects (e.g., gender, ethnicity). Nevertheless, GPT-4 was able to match or surpass the accuracy of human forecasts.

### A.2.2. FINE-TUNING AN LLM ON HUMAN DATA

Recent work has developed more sophisticated methods that may better reflect the capabilities of modern LLMs. One approach is to fine-tune an LLM with social data to embed new knowledge of human behavior or allow existing knowledge to be more easily utilized for simulation. Binz et al. (2024) build Centaur, a copy of Llama-3.1-70B-Instruct fine-tuned with data from 160 experiments. Using training-test splits, they find that fine-tuning consistently leads to more accurate simulation, though this ranges

---

[1] https://tessexperiments.org/

widely from just under 10% of the variation in the human data to over 90% for some tasks. Binz and colleagues position Centaur as a "foundation model of human cognition" and even show that fine-tuning improves alignment between model activations and human fMRI scans, and they propose future work to translate this model into a "unified theory of human cognition."

### A.2.3. INTERVIEW-BASED PROMPT VARIATION

Park et al. (2024a) conducted two-hour interviews with 1,052 participants, prepending them one at a time to the study information to create prompts for GPT-4o that would simulate each participant. To evaluate this method, they conducted an extensive battery of popular social science studies with each participant twice, two weeks apart. The difference between the human participant's responses over the two weeks was used as the ceiling for simulation accuracy. Most strikingly, they found that the simulations were 85% as accurate in predicting responses to the General Social Survey[2] as participants' first test results were at predicting their second test results, though the appropriate baseline is unclear because general knowledge—knowing the most common answers to typical survey questions, such as demographics—can perform much higher than random chance.

### A.2.4. STUDIES THAT TESTED SIMULATIONS BY PREDICTING NOVEL DATA

We believe testing simulations on predictions of novel data is the most direct way to test their ability to generalize OOD. We identified three studies that have done so already:

- Kozlowski et al. (2024) use GPT-3, with a training data cut-off date of October 2019, to simulate U.S. political polarization surrounding the COVID-19 pandemic. By using a model with training data restricted to a certain time period, they show that GPT-3-simulated liberals and conservatives came up with the same views that emerged in reality on policies such as vaccine mandates, mask requirements, and lockdowns.

- Brand et al. (2024) tested OOD performance in terms of willingness to pay for new product categories and product features (e.g., unusual flavors of toothpaste that are not available for purchase), finding low performance of LLM simulations with and without fine-tuning on in-distribution survey data (e.g., willingness to pay for common toothpaste flavors).

- Hewitt et al. (2024) conducted a large-scale test of LLM simulations. They used human data from a

database of 70 preregistered, survey-based experiments. On their correlation measure adjusted for sampling error, GPT-4 predictions correlated strongly with the average treatment effect ($r = 0.91$) with slightly higher correlation ($r = 0.94$) for a database of unpublished studies that could not have been in the LLM training data.

One study, Zhang et al. (2025a), was not in our primary scope but was focused on opinion extrapolation, which they define as "the task of predicting people's opinions on a set of new topics from their opinions on a given set of topics." They show that LLM performance on this can be increased through model-guided rejection sampling.

## B. Ethics

We believe that there is more agreement than it seems on the normative question of whether LLMs should be used for social science and AI training data. As we have detailed, proponents of LLM simulations have argued for its positive impact as an accessible and scalable data source to further social science and support training new AI systems. Most of the critiques we have seen of LLM simulations do so on empirical grounds, such as the issues of diversity and bias that we address in this position paper.

Importantly, the normative arguments that would apply even if accurate LLM simulations were developed, such as the inherent value of participatory research, tend to be presented against "entirely turning to technological solutions" (Agnew et al., 2024), which is a proposal we have not seen made in the literature. Instead, Gerosa et al. (2024) present a disclaimer that we believe most simulation researchers would endorse (italics in original): *"we neither believe nor desire for AI to completely replace human subjects in software engineering research."* We expect this would be agreed upon by most simulation researchers.

Nonetheless, like most technologies, social simulations can be used for benefit or harm (Klein & Hassabes, 2023). Researchers should account for the potential of more accurate simulations to increase risks of LLM misuse, such as spreading political messages that appear to come from real humans (Barman et al., 2024), and unintentional harm, such as inaccurate research results that spread as misinformation (Kumar et al., 2023) and reduced opportunities for financial compensation by participating in human subjects research.

## C. Limitations of Our Work

The literature on LLM simulations has only recently emerged and is scattered across disciplines and venues with varied terms and framings, so we were unable to iden-

---

[2]https://gss.norc.org/

tify work on the topic through conventional literature review methods (e.g., querying Scopus or Web of Science databases). We expect that we identified a majority of preprints and publications on the topic, given the consistency with which the authors have tended to become aware of new work, but this remains a concern—particularly for work that would be bibliometrically cut off from the aforementioned papers, such as if those unknown papers have not cited these works, if they are in different scholarly venues, or if they appear in non-English languages.

It is also difficult to draw a clear line between this topic and related topics, such as using LLMs for text annotation. For example, Veselovsky et al. (2023a) explicitly focus on the distributional accuracy of LLM-generated data relative to human data. The human data were from Abu Farha et al. (2022), social media posts made by crowdworkers that they self-identified as "sarcastic." While Veselovsky et al. (2023a) have clearly compared LLM-generated and human-generated data, the data itself is more used for natural language processing research than for social science research with human subjects, so it has clear but limited relevance to our work. Likewise, our focus has been the simulation of humans in isolation, such as a survey context, but LLMs can also be used for multi-agent and multi-turn contexts (e.g., Park et al., 2023; Piao et al., 2025; Zhou et al., 2023). We maintain this focus in part because the success of more complex simulations will likely rely on the ability to simulate each human response in isolation.

The recent development of LLMs, particularly in their application to social simulation, limits our confidence in our conclusions. Because of the fast pace of research and societal transformation from AI, such work is necessary, but it should be revisited over time as new AI system architectures emerge and become popularized.

