# OpenReview forum: "Position: LLM Social Simulations Are a Promising Research Method"
_ICML.cc/2025/Position_Paper_Track — ICML 2025 Position Paper Track poster_

### Official Review · Reviewer_966i · 2025-02-21

**Significance:** 3
**Argument Clarity:** 3
**Rating:** 3
**Confidence:** 4

**Questions:**

It would be interesting to know if the authors view this set of challenges as a comprehensive summary of the field, or if they acknowledge the potential existence of other significant hurdles. A discussion on whether a unified view on the challenges exists could provide additional clarity and depth to their analysis, making it an appealing "icing on the cake."

**Discussion Potential:**

3

**Paper Summary:**

The position paper advocates that LLM-based social simulations are promising by addressing five key challenges: diversity, bias, sycophancy, alienness, and generalization. The authors also suggest that these identified challenges can be tackled through methods such as improved prompting, fine-tuning, and complementary techniques like steering vectors and token sampling. Additionally, the paper discusses long-term strategies, including the development of conceptual models and iterative evaluation, to ensure that LLM simulations evolve alongside AI advancements. It concludes that despite significant challenges, LLM social simulations hold promise for accelerating social science research and generating humanlike synthetic data.

## update after rebuttal

I thank the authors for their efforts during the rebuttal period, and my concerns from my initial review have been addressed; therefore, I vote for acceptance.

**Position:**

Yes

**Position In Title:**

Yes

**Related Work:**

3

**Strengths And Weaknesses:**

**Strength**
1. The paper exhibits a smooth flow throughout.

2. The contributions of the discussion is significant. To me, the contributions are mainly two fold: it identifies five main challenges for current LLM-based simulations; second, it summaries and discusses some very recent works that offer promising directions for future research.

**Weaknesses**
1. A major concern is the mismatch between the identified challenges and the proposed solutions. While Table 1 effectively summarizes the connections, Section 4 could elaborate on these relationships more clearly.

2. In Section 4.5.1, the concept of "Conceptual Models" is introduced without proper sourcing or definition. The current description is overly general and lacks technical specificity.

**Support:**

3

---

> ### Author Rebuttal · Authors · 2025-03-28
>
> We are glad the reviewer sees the paper as smoothly flowing with significant contributions.
>
> > **W1**. A major concern is the mismatch between the identified challenges and the proposed solutions. While Table 1 effectively summarizes the connections, Section 4...
>
> Thanks for raising this concern (which is similar to **W2** discussed by Reviewer A9dy). Originally we wrote the paper with only the five challenges and enumerated the promising directions within each of those five sections (i.e., one challenge followed by all the proposed solutions). The main issue with that format was repetition because several methods address multiple issues at once (e.g., interview-based prompting appears to reduce bias and increase diversity, as shown by Park et al. 2024). Also, beta readers found that format confusing.
>
> We may have swung the pendulum too far in the other direction, so we will ensure these connections are clear in Section 4. For example, we think steering vectors could help with diversity (see the reference we provided in response to Reviewer S45z: openreview.net/forum?id=rwqShzb9li), but we do not expect them to be as useful with sycophancy because it seems unlikely that there is a simple sycophancy representation within the model that can be easily adjusted.
>
> We particularly appreciate the idea of elaborating on Table 1. We like having Table 1 on the first page to summarize the paper, but right now it is only referenced that single time. We have added explicit detail and references to it in Section 4 to make these connections clearer.
>
> We also aim for the new section we’ve added (detailed in our response to Reviewer S45z) to help address this mismatch by clarifying how these simulation directions/solutions would realistically be used in social science: pilots and exploratory studies, but also replications and novel studies..
>
> > **W2**. In Section 4.5.1, the concept of "Conceptual Models" is introduced without proper sourcing or definition...
>
> We agree this is overly general. Our goal here was to have a call to action for more high-level research. Unlike the specificity of prompting, steering vectors, and other established methods, this is meant to encourage foundational research in this “pre-paradigmatic” (Kuhn 1962) research area. We think there are several good examples of this already in ML/NLP more broadly (e.g., Lake et al., 2017, Rahwan et al., 2019, Shanahan et al. 2022; 2023). I find Shanahan et al.’s notion of “role play” particularly insightful, where they say, “as the conversation proceeds, the dialogue agent maintains a superposition of simulacra that are consistent with the preceding context.” That idea of what kind of personas, if any, and how many, seems to have room for exploration in the pre-paradigmatic simulation context.
>
> We are also making the paper more specific by focusing on those examples and discussing models of OOD generalization, which has been a very useful high-level framing for many LLM issues (e.g., the tendency to pattern-match and fail on slight variations of benchmarks or riddles), and OOD has been developed much more in technical detail than “role play.” We will be clearer that, even though most conceptual models currently lack technical specificity, developing that technical specificity in the simulation context should be a priority.
>
> > **Question**: It would be interesting to know if the authors view this set of challenges as a comprehensive summary...
>
> We view the five challenges as comprehensive. The most compelling other challenge we strongly considered was validation (How do we verify that the simulation is accurate, particularly if we lack human data?). We decided that validation was better conceptualized in this paper as a component of each challenge (e.g., How do you validate unbiasedness? How do you validate diversity?).
>
> We appreciate the consideration of a “unified view.” Actually, before we even landed on the framing of multiple challenges, the paper was going to be about why OOD generalization should be the priority for researchers. The idea there would be that each other challenge can be presented as OOD generalization (e.g., bias as failing to generalize to real groups who face systemic bias, diversity as ensuring that results aren’t confined to a narrow distribution of real people).
>
> In the new section on social science applications (figure sketch: https://imgur.com/a/0PQAqlP), we have included a paragraph on this topic. In particular, all of these applications have a central goal: accuracy (as is the goal of most ML/NLP domains). We see the five challenges as important ways in which simulations fail to be accurate; this is meant to be reflected in the “inaccuracies” phrasing in Table 1. We will ensure this is clear, including the elaboration on Table 1 in Section 4, and we will qualify it that this accuracy needs to be verifiable (i.e., we know it is accurate) and ethical (e.g., ensures that the public can still participate in the research process).

---

> > ### Comment · Reviewer_966i · 2025-04-05
> >
> > Thanks for the response! My concerns have been addressed. I will maintain my score since it is already a positive rating.

---

### Official Review · Reviewer_g7Sc · 2025-03-09

**Significance:** 3
**Argument Clarity:** 4
**Rating:** 4
**Confidence:** 4

**Questions:**

Please see my previous comments.

**Discussion Potential:**

3

**Paper Summary:**

This paper argues that LLMs can be productively used to generate behavioral data that would otherwise be collected from human research subjects. More specifically, LLM social simulations can be used for ``exploratory social research in which surfacing interesting possibilities can be more important than avoiding false positives" (note: I am specifically pulling this quote out of the paper to return to it in my comments). The authors conduct a thorough review of studies and commentaries across a wide variety of social sciences that use LLM social simulations, and the authors highlight 5 challenges and promising directions for research: (i) diversity -- LLMs tend to produce outputs that are more homogeneous than the variation in the human population; (ii) bias -- LLMs tend to produce systematically inaccurate outputs for particular social groups; (iii) sycophancy -- LLM outputs are over-optimized to produce outputs that receive positive feedback from users, and so it may censor human behavior that we would like to study; (iv) alienness -- LLMs match human behavior with non-human like mechanisms; and (v) generalization -- LLM sims have only been evaluated on known instruments and we do not know their performance on novel designs. The authors then highlight valuable directions for future research.

**Position:**

Yes

**Position In Title:**

Yes

**Related Work:**

4

**Strengths And Weaknesses:**

The paper articulates a clear position that is supported with a thorough review of existing research across the social sciences which uses LLM sims. The summary of papers in of itself is quite valuable.

In my view, there are several weaknesses of the position paper:

(1) I doubt the position itself will generate much additional discussion. In particular, as evidenced by the authors thorough review, there is already substantial interest in LLM social simulations across computer science and the social sciences. Furthermore, many of the directions described in the paper have been highlighted in particular papers documented in the review. Of course, it is valuable to assemble this all in one place.

(2) Aside from the general position "LLM sims are a promising research method," the only qualification added is the specific caveat that LLMs can now be used for "exploratory social research." I am not quite sure what is the end goal of this position. In particular, should the community be tackling these five open challenges so that we can eventually replace human subjects in social science experiments --- as an example, rather than studying psychology by recruiting human subjects, we should instead test for biases in LLMs? Or is it to remain as a tool to cheaply pilot studies that we then plan to scale on human subjects? To the authors, of this position are those use cases the same or are they different? If they are different, do we measure progress on the 5 challenges differently?

(3) It is not apparent to me why alienness is a general problem for using LLM sims. In particular, consider surveys used to study political opinions or surveys used to measure various economic outcomes. In these contexts, we clearly do not care about the process by which a person arrived at the answer but we only care about the answer. So why should we care about the specific "mechanism" of an LLM? At minimum, alienness is a challenge for only specific use cases of LLM sims, and it would be valuable for the authors to clarify what those specific use cases are.

(4) Generalization appears to be the most important direction for future work -- especially given our own inability to accurately generalize about LLM performance in other domains (e.g., [1]). Embedding LLM sims into the research process in the social sciences poses significant risks if we cannot reliably predict how well LLMs replicate human subjects on novel designs. If anything, I think the authors need to invest more time in describing why this challenge is particularly important.

[1] Vafa et al. "Do large language models perform the way people expect? measuring the human generalization function" ICML 2024.

**Support:**

4

---

> ### Author Rebuttal · Authors · 2025-03-28
>
> We appreciate that the reviewer sees the paper as clear and the synthesis as valuable.
>
> > (1) I doubt the position itself will generate much additional discussion. In particular, as evidenced by the authors thorough review, there is already substantial interest in LLM social simulations...
>
> There has been a lot of discussion and interest, but we’d respectfully note that there were only 36 preprints or publications empirically running LLM social simulations as of January 5th (and a handful since then). While there has been excitement and “hype,” this is actually a relatively small amount of literature compared to most LLM applications. There has been very little uptake in social science. The social science papers we highlight are all focused on methodology itself, rather than utilizing the methodology for typical social science objectives. We have clarified this in the paper: the field has a lot of room to grow and work to be done, even though we see it as promising.
>
> > Furthermore, many of the directions described in the paper have been highlighted in particular papers documented in the review. Of course, it is valuable to assemble this all in one place.
>
> We’re glad the reviewer agrees with the value of assembly, which we also see as the primary contribution: our aim was to write a position paper that provides “structure and clarity” to this nascent and somewhat chaotic new research area. We agree that not many directions are novel (and, indeed, none are novel in terms of LLM research outside of the simulation domain). We have incorporated this into the limitations section.
>
> > (2) Aside from the general position "LLM sims are a promising research method," the only qualification added is the specific caveat that LLMs can now be used for "exploratory social research."...
>
> This is a great point. We don’t aim for simulations to completely replace human subjects. We previously emphasized this aim more in the draft but cut it due to space. (We still emphasize “complementing human data” rather than replacement in the introduction). However, we also don’t expect simulations to remain a tool to cheaply pilot studies.
>
> Yes, these use cases are different, and based on all the reviewers’ feedback, we are using the 9th page to add a new section on six specific applications: pilots, exploratory studies, exact replications, sensitivity analysis, novel studies when human subjects research is difficult, and novel studies when human subjects research is impossible. We provide more detail in our first response (Reviewer S45z, figure sketch: https://imgur.com/a/0PQAqlP), and in terms of measuring them differently, we will emphasize the common metric of accuracy across the five challenges (discussed in the last paragraph in our response to Reviewer 966i). We welcome further suggestions in this direction, particularly changes to the paper that can motivate more ML researchers to work in this area.
>
> > (3) It is not apparent to me why alienness is a general problem for using LLM sims...
>
> I think we agree with the reviewer here. “Alienness” and “generalization” are two distinct issues. In alienness, the LLM sim is inaccurate when you zoom in, and generalization is an issue when you zoom out. These are similar, and one could view both as generalization issues, but we found it more useful to distinguish these two after thinking through the different issues faced in social science, such as the Big Five example for alienness. We had put some of the generalization examples into the appendix, particularly A.2.4 (the three examples of generalization simulation testing we identified), so we can prioritize that in the camera-ready main text.
>
> This point may also be partly addressed by the change we discuss in response to Reviewer 966i, where we will more emphasize the unifying goal of accuracy. So, yes, if the low-level mechanisms of a social phenomena do not matter for the research, then we would not see alienness as a challenge/goal. However, given the limited reliability of simulations, we could have more confidence in their results if we also verified the low-level mechanisms. We will ensure the paper reflects this nuance, as well as that some social science paradigms (e.g., cognitive psychology) care a lot more about mechanisms than others (e.g., macroeconomics).
>
> > (4) Generalization appears to be the most important direction for future work...
>
> We appreciate and agree with this suggestion. As also discussed in response to Reviewer 966i, we think generalization is a fundamental challenge across many LLM applications, and we will emphasize its importance in the new section on social science applications.
>
> We will also include a citation to Vafa et al. It will be important to consider that misalignment in generalization function as more simulation studies are published and we need to assess the evidential strength of LLM simulation reliably performing in one type of human behavior to reliable performance in terms of different behavior.

---

### Official Review · Reviewer_A9dy · 2025-03-13

**Significance:** 3
**Argument Clarity:** 3
**Rating:** 3
**Confidence:** 3

**Questions:**

no

**Discussion Potential:**

2

**Paper Summary:**

This paper focuses on using LLMs for social simulation. The authors summarize and point out five key challenges: diversity, bias, sycophantsancy, alienness, and generalization. They position that by addressing these challenges, it would be promising to using LLMs for social simulation.

**Position:**

Yes

**Position In Title:**

Yes

**Related Work:**

3

**Strengths And Weaknesses:**

## Strengths
1. This paper is well-structured and is easy to follow.
2. The scope of this paper is clearly and defined.
3. This paper points out some interesting claims and insights, which would be useful for researchers in this area.

## Weaknesses
1. Literature review may be not sufficient. The authors claimed "Sycophancy was not explicitly discussed in papers we reviewed", however, as far as I know, there are already existing literature on "sycophancy" or "flattering" [1,2,3]
2. Humanlikeness of LLMs is the foundation of LLM-based social simulation. However, this paper focuses on pointing out several challenges when using LLMs for simulation, making readers (e.g., me) think that LLMs are acturally not humanlike enough. This somehow contradicts with the main position of this paper: LLM Social Simulations Are a Promising Research Method. That is, I thought that I would see more evidence that LLMs are humanlike.
3. There are some confusions. In the introduction, the authors hightlight that "In this position paper, we show the promise of LLM
social simulations by identifying five key tractable challenges and promising directions for future research.". I think that "identifying challenges" does not sounds like "promising".

[1] Sycophancy in Large Language Models: Causes and Mitigations

[2] Flattering to Deceive: The Impact of Sycophantic Behavior on User Trust in Large Language Model

[3] Towards Understanding Sycophancy in Language Models

[4] Sycophancy to subterfuge: Investigating reward tampering in language models,

**Support:**

3

---

> ### Author Rebuttal · Authors · 2025-03-28
>
> We are glad the reviewer sees the paper as well-structured, well-scoped, and having interesting claims and insights.
>
> > **W1**. Literature review may be not sufficient. The authors claimed "Sycophancy was not explicitly discussed in papers we reviewed", however, as far as I know, there are already existing literature on "sycophancy" or "flattering" [1,2,3]
>
> We apologize for the unclear wording. We have changed the way this is phrased to be clear that sycophancy was not explicitly discussed **in the context of LLM social simulations**. We appreciate the four citations and have included them in our discussion (note that we already had a reference to [2] as a review of the broader sycophancy literature).
>
> We will also ensure that this section covers the ways in which the sycophancy literature could contribute to simulations, such as if fine-tuning and preference tuning are done in ways that reduce sycophancy—making those models better for simulations. We welcome further suggestions for content or clarity.
>
> > **W2**. Humanlikeness of LLMs is the foundation of LLM-based social simulation…
>
> We think they are promising for three reasons: (i) general LLM capabilities, (ii) simulation results to date that show limited, but significant, accuracy, and (iii) the challenges we articulate (which we will discuss in response to **W3**).
>
> ### General LLM capabilities
>
> We take it as a generally established finding that LLMs have become more and more humanlike, leading to generally increased task performance, such as on established benchmarks for scientific question-answering (GPQA), mathematics (MATH), and software engineering (SWE-bench Verified). We will add a brief summary of these general capabilities to the paper.
>
> ### Simulation results to date
>
> There are 3 examples of existing research that we think are particularly promising. These were in the appendix, but we will move more detail to the main text:
>
> 1. Hewitt et al. (2024), which simulates a range of preregistered, nationally representative survey experiments with GPT-4, with an average disattenuated correlation of 0.85, which persists with unpublished studies that could not be in the LLM training data.
> 2. Binz et al. (2024), which shows that fine-tuning Llama-3.1-70B-Instruct on data from 160 experiments increases accuracy in that context, sometimes an over 90% increase.
> 3. Park et al. (2024), which shows that prompting GPT-4o with interview transcripts from each person in a nationally representative sample leads to 85% accuracy, relative to people replicating their own responses, on the General Social Survey (https://gss.norc.org/).
>
> We will add discussion of how these advances are even more impressive when one considers how recently the methodology of LLM social simulation was developed. For example, Park et al. (2024) was the first paper to use interview transcripts, so there are likely many low-hanging fruit for improving results further (e.g., selecting the most informative interview questions, combining with other methods like the fine-tuning of Binz et al. (2024) and more recent work like SubPOP: https://arxiv.org/abs/2502.16761), which fine-tunes on distributions of subpopulations, supporting the “Distribution Elicitation” argument in our paper.
>
> It is impressive that these results have been achieved even with the obvious nonhumanlikeness of current LLMs, such as limited general-purpose capabilities and being built on nonhumanlike training data, such as low-quality internet text and software code. Once models specifically trained and tuned at scale for simulation, performance may greatly increase.
>
> ### Challenges as a reason for optimism (response to **W2** and **W3**)
>
> > **W3**. There are some confusions…
>
> We appreciate the reviewer pointing out this confusion in the way we framed the paper with the focus on “identifying challenges.” We meant to make an argument that, by outlining these challenges and showing that they are sufficiently tractable, we show the promise of LLM social simulations. If we identified challenges but they all seemed very difficult and without any promising directions, that would be evidence that the field is **not** promising.
>
> In the case of bias, diversity, and sycophancy, we think there are solutions with prompting (e.g., interview transcripts), steering, token sampling, and tuning and training. In the case of alienness and generalization, progress will be more difficult, but we think we present a relatively clear encapsulation of the challenges that opens up opportunities for future work in the direction of conceptual modeling and iterative evaluation.
>
> We didn’t communicate this clearly enough in the submission, so we have reworded the paper to make this clear from the beginning why exactly our literature review suggests it is a promising research method. The new section on six social science applications we discuss in our response to Reviewer S45z (figure sketch: https://imgur.com/a/0PQAqlP) may also help address this.

---

> > ### Comment · Reviewer_A9dy · 2025-04-03
> >
> > Thanks for the responses. I would reconsider my rating.
> >
> > BTW, the following related works may be helpful:
> > 1. SOTOPIA: Interactive Evaluation for Social Intelligence in Language Agents, ICLR 2024
> > 2. Self-Alignment of Large Language Models via Monopolylogue-based Social Scene Simulation, ICML 2024
> > 3. Scaling Synthetic Data Creation with 1000000000 Personas

---

> > > ### Author Response · Authors · 2025-04-03
> > >
> > > We appreciate the 3 related works, which we will also include.
> > >
> > > We are glad to hear you would reconsider your rating. If there are still concerns you have that motivate a 2 (Weak reject) rating, we would appreciate clarification of what those are so we may be able to address them. If your concerns have been addressed, we would be grateful for a revision of your score. Thank you.

---

### Official Review · Reviewer_S45z · 2025-03-14

**Significance:** 3
**Argument Clarity:** 2
**Rating:** 3
**Confidence:** 4

**Questions:**

Since this paper primarily surveys literature in computer science, have the authors also considered research from other fields, such as sociology, psychology, and education, that explore the use of LLMs?

**Discussion Potential:**

2

**Paper Summary:**

This position paper examines the potential of LLM social simulations for studying human behavior and training AI systems. While adoption by social scientists remains limited, the authors identify five key challenges that, if addressed, could unlock their promise. Drawing from empirical studies and commentary, they highlight advances in prompting, fine-tuning, and complementary methods as promising solutions. They argue that LLM simulations are already useful for exploratory research in psychology, economics, sociology, and marketing, and with ongoing AI advancements, developing conceptual models and evaluation frameworks should be a research priority.

**Position:**

Yes

**Position In Title:**

Yes

**Related Work:**

2

**Strengths And Weaknesses:**

This position paper proposes five tractable challenges, which I find important and largely agree with. However, the discussion on promising directions feels somewhat vague and lacks significant impact.

A minor suggestion: the paper is quite dense with text. Adding a graph or illustration could improve readability and make the key points more accessible.

**Support:**

3

---

> ### Author Rebuttal · Authors · 2025-03-28
>
> We appreciate the reviewer’s support of the paper.
>
> > However, the discussion on promising directions feels somewhat vague and lacks significant impact.
>
> We think we can address this with the extra page for the camera-ready. Our goal with the promising directions section was to outline a broad toolkit of methods applicable to this context. Our hope is that a reader might say, “Ah, nobody’s used steering vectors in this domain yet? I know how to do that. I should try it.” For example, there’s a forthcoming ICLR paper (https://openreview.net/forum?id=rwqShzb9li) that identifies a vector for left-right political perspective, which seems promising for the simulation context (e.g., testing counterfactuals).
>
> Because we want to maintain that section as that wide-ranging resource, we have used the 9th page to add a section following promising directions on specific applications, where we will stake out a more opinionated, speculative view on six that we expect to see.This speculation requires qualification, e.g., these applications are only in approximate order of increasing difficulty, so it is possible that later applications could be successfully implemented before the former. We have sketched a possible figure for this (https://imgur.com/a/lzqpYV6).
>
> 1. **Pilots**: LLMs are used to generate and refine ideas (e.g., narrow a list of 100 possible experimental treatments to 5), but it is followed up with a real study with human subjects as the main result;
> 2. **Exploratory studies**: LLMs are used for an entire study, but the research is intended as exploratory rather than confirmatory. For example, a study could aim to identify many ways in which political orientation can manifest in unstructured text, rather than to confirm any hypothesis about their relative frequency or causal relationships.
> 3. **Exact replication**: LLM simulations are run with the same methodology as a human subjects study. Either the LLM data is combined with the human data, with conservative assumptions, to increase statistical power (e.g., using the “mixed subjects design”: https://papers.ssrn.com/sol3/papers.cfm?abstract_id=5133034), or the LLM data is tested separately but only to increase or decrease confidence in the primary human subjects finding.
> 4. **Sensitivity analysis**: LLMs simulations are first confirmed to verify human subjects findings but then implemented with minor variations. For example, the LLMs could be tested with many different paraphrasings of a survey question to provide some evidence of whether the human subjects results may depend on a particular phrasing. This could include variations in the population, such as using LLMs with analogous fine-tuning to US and UK survey data to assess if results would change.
> 5. **Complete studies when human subjects research is possible but impractical**: For example, LLMs could help make original research more accessible to students or researchers in low-income countries who have limited funding. At this point, LLMs could be used for the entire study, but ideally the results would still be later replicated with a new human subjects study—as is also the case for human subjects research!
> 6. **Complete studies when human subjects research is impossible**: In cases such as a test that is unethical to do with real humans or a large-scale government policy change. In these cases, direct human subjects validation is not possible, so the LLM simulation needs to be most reliable. As many local components that can be validated with humans still should be (e.g., the behavior of one individual in response to one facet of the policy change), and validation of related behaviors can be used.
>
> > A minor suggestion: the paper is quite dense with text. Adding a graph or illustration could improve readability...
>
> We have sketched a figure with the six applications (https://imgur.com/a/0PQAqlP), but we welcome suggestions for modification or different figures!
>
> > Since this paper primarily surveys literature in computer science, have the authors also considered research from other fields, such as sociology, psychology, and education, that explore the use of LLMs?
>
> Yes, as far as we can tell, we’re aware of most studies on the topic, across any field. Of the 36 papers in Table A1, roughly 12 papers are primarily authored by social scientists, and 2 are highly interdisciplinary with the other 22 being primarily computer scientists. Table A2 has 4 social science venues and two CHI (human-computer interaction) papers. There are also a few papers that have come out since our initial submission that we have added (e.g., Suh et al. arxiv.org/abs/2502.16761).
>
> The new section on applications also engages with the latest methods in those fields (e.g., recent open science, including not just preregistration, replication, etc. but recent initiatives like clearly defined estimands: doi.org/10.1177/00031224211004187). This also includes infrastructure like public datasets and benchmarks.

---

### Decision · Program_Chairs · 2025-04-30

**Decision:**

Accept (poster)

**Comment:**

The paper addresses an important, interesting, and timely issue.
It provides a very nice overview of the field. While hype around LLMs is high, the focus on social science makes the paper a lot more focused and an interesting discussion point.

The author promised a number of modifications to make the paper more concrete, to add missing discussions and references, clear up some misunderstandings, and to make it more pleasurable to read (improved flow and connections, illustrations & figures). The replies are detailed providing a clear indication of what will be changed, and those will address the concerns well. Overall and excellent rebuttal - with the only remaining concern on how the authors will manage to squeeze all the promised additional material in the 1 extra page.